# A radical *S*-adenosyl-L-methionine enzyme and a methyltransferase catalyze cyclopropane formation in natural product biosynthesis

Wen-Bing Jin [1], Sheng Wu[1], Xiao-Hong Jian[1], Hua Yuan [1] & Gong-Li Tang [1]

Cyclopropanation of unactivated olefinic bonds via addition of a reactive one-carbon species is well developed in synthetic chemistry, whereas natural cyclopropane biosynthesis employing this strategy is very limited. Here, we identify a two-component cyclopropanase system, composed of a HemN-like radical *S*-adenosyl-L-methionine (SAM) enzyme C10P and a methyltransferase C10Q, catalyzes chemically challenging cyclopropanation in the anti-tumor antibiotic CC-1065 biosynthesis. C10P uses its [4Fe-4S] cluster for reductive cleavage of the first SAM to yield a highly reactive 5'-deoxyadenosyl radical, which abstracts a hydrogen from the second SAM to produce a SAM methylene radical that adds to an $sp^2$-hybridized carbon of substrate to form a SAM-substrate adduct. C10Q converts this adduct to CC-1065 via an intramolecular $S_N2$ cyclization mechanism with elimination of *S*-adenosylhomocysteine. This cyclopropanation strategy not only expands the enzymatic reactions catalyzed by the radical SAM enzymes and methyltransferases, but also sheds light on previously unnoticed aspects of the versatile SAM-based biochemistry.

---

[1] State Key Laboratory of Bio-organic and Natural Products Chemistry, Center for Excellence in Molecular Synthesis, Shanghai Institute of Organic Chemistry, University of Chinese Academy of Sciences, Chinese Academy of Sciences, 345 Lingling Road, Shanghai 200032, China. These authors contributed equally: Wen-Bing Jin, Sheng Wu. Correspondence and requests for materials should be addressed to H.Y. (email: huayuan@sioc.ac.cn) or to G.-L.T. (email: gltang@sioc.ac.cn)

Cyclopropane moiety-containing complex chemical molecules present significant challenges for synthetic chemists, while the frequent appearance of the cyclopropane moiety in preclinical/clinical drug molecules has spurred significant advances in elegant synthetic methods[1–3]. For example, cyclopropanation of unactivated olefinic bonds using a reactive one-carbon species is well developed in synthetic chemistry[3]. In biology, S-adenosyl-L-methionine (SAM) often serves as a reactive one-carbon donor. Hence according to the degree of dependence on SAM, enzymatic cyclopropanation can be divided into three classes in natural product biosynthesis (Supplementary Fig. 1). The first class are those reactions that are independent of SAM, including (i) formation of cyclopropane-containing terpenoids via inter- and intramolecular electrophilic addition[4]; (ii) biosynthesis of the alkaloid cycloclavine through an α-ketoglutarate-dependent, non-heme iron oxygenase EasH-catalyzed oxidative rearrangement[5]; and (iii) construction of cyclopropane-containing building blocks for nonribosomal peptides and hybrid nonribosomal peptide-polyketide compounds using halogenated carrier protein-linked intermediates as the substrates for $S_N2$-like cyclopropanation[6–9]. The second class include those that only require the activated methyl group of SAM, e.g., formation of cyclopropane fatty acids by cyclopropane fatty acid/mycolic acid synthases that catalyze direct transfer of the reactive one-carbon species from SAM to double bonds involving a mechanism of carbocationic intermediates (or transition states)[10,11]. The last class are those that use SAM to supply all the three-carbon source for the cyclopropane moiety, e.g., biosynthesis of 1-aminocyclopropane-1-carboxylate as a precursor to the plant hormone ethylene, and the cyclopropane warhead of colibactin through a carbanion-induced intramolecular $S_N2$ reaction mechanism with elimination of methylthioadenosine[12,13]. Recently, engineered cytochrome P450 enzymes were reported to have the ability to catalyze olefin cyclopropanation via carbene transfer[14], which strongly hints that more hidden chemistries, including cyclopropanation reactions, remain to be discovered in the enzyme catalyst toolbox[15].

CC-1065 (**1**) belongs to the spirocyclopropylcyclohexadienone family of natural products. This family also includes gilvusmycin (**2**), yatakemycin (YTM, **3**), duocarmycin SA (**4**), and duocarmycin A (**5**) (Fig. 1)[16–21]. All these chemical molecules share a highly reactive cyclopropane moiety, which serves as a warhead and endows these compounds with exceptionally potent cytotoxicity via a unique shape-dependent DNA alkylation mechanism[22,23]. Thus, these compounds have attracted continuing concerns for chemical, biological, and pharmaceutical studies[24,25]. Currently, a series of analogues of **1** and the duocarmycins have been chemically developed for the use in targeted tumor therapies with two members (SYD985 and MDX-1203) into Phase I clinical trials[26–28].

Our group has focused on the biosynthetic studies of **1** and **3**, and we have identified their biosynthetic gene clusters (BGCs), unraveled a unified biosynthetic origin of the benzodipyrrole subunits in **1**, and proposed their biosynthetic pathways,

respectively[29,30]. Gene inactivation and biosynthetic intermediate characterization confirmed that a HemN-like radical SAM enzyme C10P was involved in the formation of the cyclopropane moiety in **1**[29], but how the cyclopropane formation proceeds is still obscure. Here, we show that a methyltransferase C10Q is also required for the cyclopropanation process. Moreover, we establish that a two-component cyclopropanase system, composed of C10P and C10Q, is able to catalyze the cyclopropyl moiety formation in **1** biosynthesis. Product analysis and labeling experiments support that reductive cleavage of the first molecule of SAM ($SAM_1$) from C10P yields a highly reactive 5′-deoxyadenosyl (dAdo) radical, which abstracts a hydrogen atom from the activated methyl group of the second molecule of SAM ($SAM_2$) from C10P. A SAM methylene radical is thus produced and then adds to an $sp^2$ carbon (C-11) of a pyrrole moiety in the substrate **6** to form a SAM-substrate covalent adduct. Finally, this adduct is biotransformed by C10Q to **1** via an intramolecular $S_N2$ cyclization mechanism with S-adenosylhomocysteine (SAH) as a co-product. The findings thereby represent a distinct naturally occurring cyclopropanation strategy and also expand the enzymatic reactions catalyzed by the radical SAM enzymes and methyltransferases.

## Results

**Characterization of a two-component cyclopropanase system.** We have previously showed that the radical SAM enzyme C10P is essential for the formation of the cyclopropane moiety in **1** (Fig. 2d), and inactivation of *c10P* produced a compound **6** (Fig. 2i)[29]. Subsequently, gene deletion mutations led us to the identification of a SAM-dependent methyltransferase gene *c10Q* that is also required for the production of **1** (Fig. 2e and Supplementary Fig. 2). Intriguingly, this Δc10Q mutant bears the same metabolite profile as that of the Δc10P mutant, that is, mutation of *c10Q* also led to accumulation of the compound **6** in the fermentation broth (Fig. 2e). Moreover, introduction of the *c10Q* gene in trans into the Δc10Q mutant restored the biosynthesis of **1** (Fig. 2f). These genetic experiment results inspired us to surmise that C10P and C10Q may work together to convert the compound **6** to **1**.

To test this hypothesis, we first expressed *c10P* and *c10Q* in E. coli, and purified the recombinant enzymes as $His_6$-tagged fusion proteins, respectively (Supplementary Fig. 3). Despite many efforts we could not obtain the C10P proteins with high purity. Fortunately, using the C10P sequence as a query we retrieved a cryptic BGC from Shewanella woodyi ATCC 51908 (GenBank accession number NC_010506) that shows high homology and synergy with that of **1** (Supplementary Fig. 4). We then cloned and integrated the *Swoo_2002* gene (the *c10P* counterpart) into the chromosome of the Δc10P mutant, and found that Swoo_2002 could function as a C10P surrogate (Fig. 2g, h). Subsequent heterologous overexpression of Swoo_2002 in E. coli led to obtaining relatively pure recombinant protein (Supplementary Fig. 3). After anaerobic reconstitution, Swoo_2002

**Fig. 1** Chemical structures of the spirocyclopropylcyclohexadienone family of natural products. This family of natural products includes CC-1065 (**1**), gilvusmycin (**2**), yatakemycin (YTM, **3**), duocarmycin SA (**4**), and duocarmycin A (**5**)

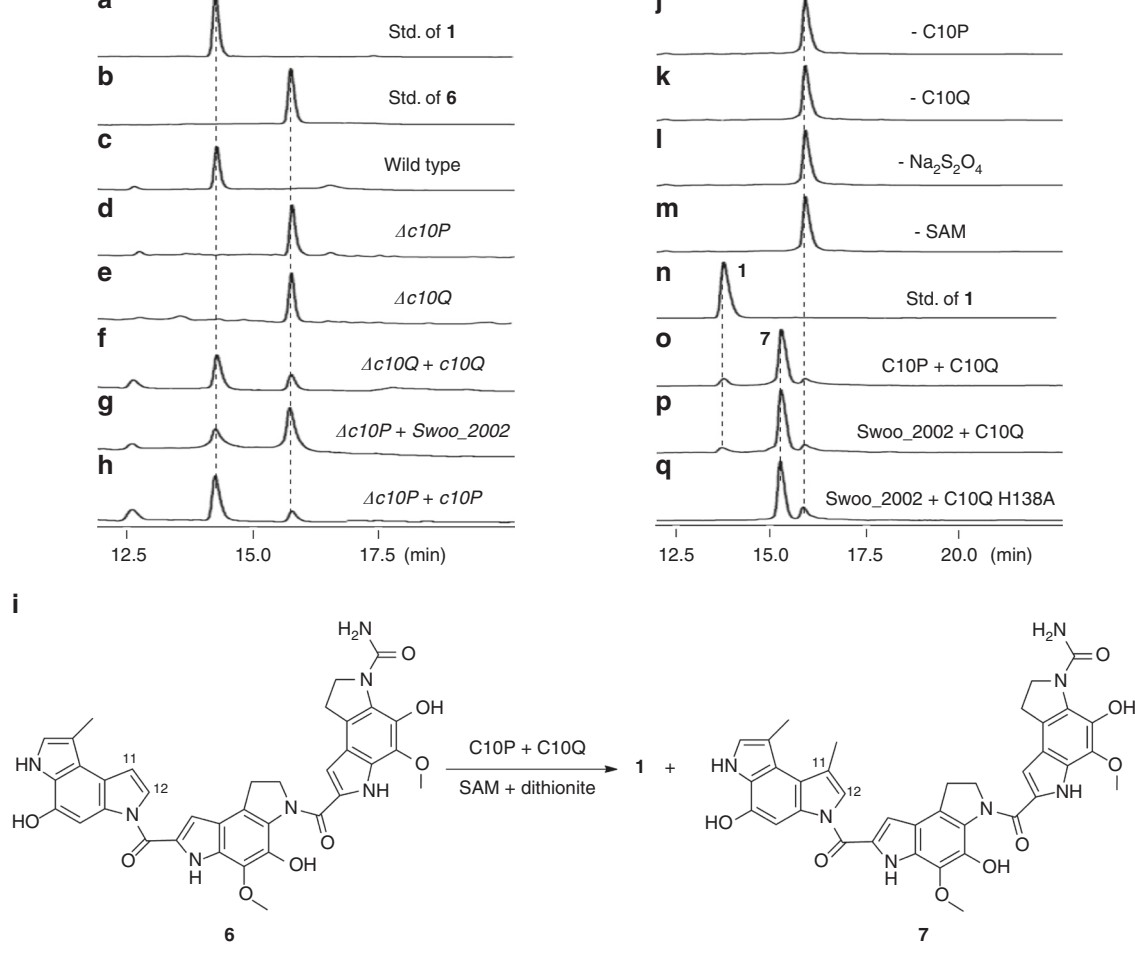

**Fig. 2** Characterization of a two-component cyclopropanase system. **a–h** HPLC analysis of the relevant metabolites (UV at 374 nm). **a** Standard of **1**.**b** Standard of **6**. **c** Wild type *Streptomyces zelensis* NRRL 11183. **d** The Δ*c10P* mutant. **e** The Δ*c10Q* mutant. **f** The Δ*c10Q* mutant complemented with the *c10Q* gene. **g** The Δ*c10P* mutant complemented with the *Swoo_2002* gene. **h** The Δ*c10P* mutant complemented with the *c10P* gene. **i** Structures of **6** and **7**. **j–q** HPLC analysis of enzymatic products. **j–m** are the control reactions without C10P, C10Q, Na$_2$S$_2$O$_4$, and SAM, respectively. **n** Standard of **1**. **o** The complete reaction for C10P and C10Q. **p** The complete reaction for Swoo_2002 and C10Q. **q** The reaction for Swoo_2002 and C10Q H138A. All the in vitro enzymatic activity assays were performed in an anaerobic glove box with less than 1 ppm of O$_2$. Reactions were conducted in Tris•HCl buffer (Tris 50 mM, NaCl 100 mM, glycerol 10%, pH 8.0) with the following composition: 1 μM reconstituted Swoo_2002, 2 μM C10Q, 10 μM substrate **6**, 1 mM SAM, 5 mM DTT, 5 mM MgCl$_2$, 5 mM Na$_2$S$_2$O$_4$ and 7% DMSO. The reactions were incubated at 28 °C for 12 h

contained 4.29 ± 0.12 of iron and 4.70 ± 0.14 of sulfide per polypeptide; its UV–visible absorption spectra revealed an A280/A420 ratio of 3.4:1 and an apparent decrease of A420 upon dithionite reduction (Supplementary Fig. 3). These evidences are well consistent with the existence of one [4Fe-4S] cluster per polypeptide of HemN-like radical SAM enzymes[31,32]. As anticipated, in vitro enzymatic assays showed that only the two-component system, composed of reconstituted C10P (or reconstituted Swoo_2002) and C10Q, could catalyze the formation of **1** using the compound **6** as substrate in the present of SAM and sodium dithionite under strictly anaerobic conditions, whereas all other control experiments did not generate any unexpected products (Fig. 2j–p and Supplementary Fig. 5).

Compound **7** in the complete enzymatic reactions attracted our attention (Fig. 2o, p). Next, we isolated sufficient quantity of this compound for structural characterization from large-scale enzymatic assays. High-resolution mass spectrometry (HR-MS) and nuclear-magnetic resonance (NMR) spectroscopy revealed that **7** is a methylated derivative of **6** at the C-11 position (Fig. 2i and Supplementary Figs 6–8). The structure of **7** prompted us to question whether it is a biosynthetic intermediate. Subsequent

enzymatic assays, however, showed that **7** could not be converted to **1** (Supplementary Fig. 9), which suggests that **7** may be an off-pathway product. In agreement with this observation, **7** is only detected under the in vitro enzymatic conditions but not in any fermentation cultures of related gene deletion mutants (Fig. 2d, e). We will discuss the relationship between **7** and **1** below.

These results also indicated that the radical SAM enzyme and the methyltransferase worked very closely. Indeed, isothermal titration calorimetry (ITC) studies confirmed that there exists an obvious interaction between Swoo_2002 and C10Q ($K_D = 8.0$ μM) (Supplementary Fig. 10). Although we tried many different enzymatic reaction conditions, we could not improve the production of **1** (Supplementary Figs 11–15). Collectively, our genetic and biochemical results demonstrate that a two-component cyclopropanase system, composed of a radical SAM enzyme and a methyltransferase, catalyzes the formation of the cyclopropyl moiety in biosynthesis of **1**.

**Mechanistic studies of the HemN-like radical SAM enzyme.** Bioinformatics analysis revealed that the C10P (and Swoo_2002) protein belongs to a HemN-like coproporphyrinogen III oxidase

that is a radical SAM enzyme (Supplementary Fig. 16). Radical SAM superfamily proteins generally contain a highly conserved CxxxCxxC motif that coordinates a [4Fe-4S] cluster for binding and reductive cleavage of SAM[33]. As expected, mutagenesis experiments confirmed that the conserved cysteine motif of Swoo_2002 (C57A) is essential for the catalysis (Supplementary Fig. 17). Previous studies have revealed that there are two bound SAM molecules in the *E. coli* HemN crystal structure[32]. One molecule of SAM ($SAM_1$) juxtaposed in close proximity to the [4Fe-4S] cluster as in other radical SAM enzymes is harnessed to yield a highly reactive dAdo radical[34,35], whereas the function of the second SAM ($SAM_2$) remains elusive. Multiple sequence alignment revealed that the two SAM-binding motifs are highly conserved within HemN-like radical SAM enzymes (Supplementary Fig. 16). Subsequent mutation of both motifs from Swoo_2002 proved that they are also required for the activity (Supplementary Fig. 18). During our enzymatic cyclopropanation reactions, we simultaneously observed the formation of 5′-deoxyadensine (5′-dA) and SAH (Fig. 3a–e and Supplementary Fig. 19). We then carried out a time-course analysis of the concentration changes of **1**, **6**, **7**, 5′-dA, and SAH as the reaction proceeds. The results showed that 5′-dA and SAH were always produced at 1:1 stoichiometry and that the amount of 5′-dA (or SAH) was approximately equivalent to that of **1** plus **7** (Fig. 3f). These evidences may suggest that two molecules of SAM are consumed to yield one molecule of 5′-dA and one molecule of SAH during a single turnover with one SAM as a radical initiator and the other one as a methyl donor. This is consistent with the

two bound molecules of SAM located in the HemN-like proteins[32].

Next, we probed possible enzymatic intermediates during the cyclopropanation process from **6** to **1**. Fortunately, we were able to detect a small peak (**8**) by high performance liquid chromatography (HPLC) analysis only from the highly concentrated complete enzymatic reactions via lyophilization (Fig. 3a–e and Supplementary Fig. 20). HR-MS analysis revealed that **8** exhibited molecular ions at $m/z = 1088.3761$ ([M]$^+$) and 1110.3763 ([M-H + Na]$^+$) (Fig. 3g). Further MS/MS analysis showed that the fragment ions produced included 250.0945, 384.1216, 704.2466, and 838.2749 (Fig. 3i), all of which are well-matched with several predicted fragmentation patterns of a covalent adduct formed between the substrate **6** and SAM (Fig. 3j).

The compound **8** may be a key intermediate during cyclopropanation, suggesting that the radical SAM enzyme probably mediates an addition of a SAM methylene radical to the C-11 position of the substrate **6**. To verify this proposal, we used $CD_3$-SAM instead of SAM for the complete enzymatic assays. The produced 5′-dA had a mass increment of +1 Da ([M + H]$^+$ = 253.1084 for D-5′-dA) (Fig. 4b and Supplementary Fig. 21), consistent with the proposal that the dAdo radical generated from the first molecule of $CD_3$-SAM abstracts a deuterium atom from the methyl group of the second molecule of $CD_3$-SAM to give D-5′-dA and a SAM methylene radical. The intermediate **8** showed a mass shift of +2 Da ([M]$^+$ = 1090.3809 and [M-H + Na]$^+$ =1112.3801 for $D_2$-**8**), which further confirms

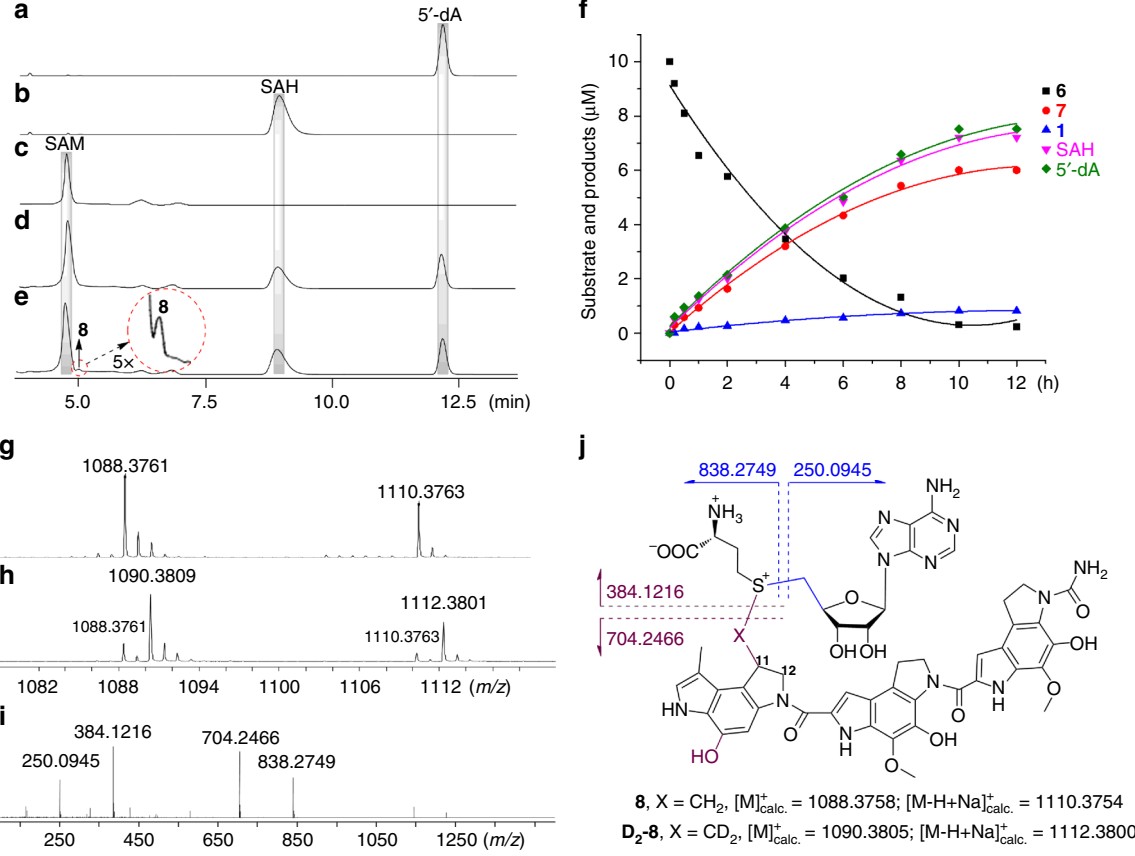

**Fig. 3** Analysis of reaction products from the cyclopropanation process. **a–e** HPLC analysis of the enzymatic products (UV at 260 nm). **a** Standard of 5′-dA. **b** Standard of SAH. **c** The control reaction using boiling-inactivated enzymes. **d**, **e** are the concentrated products from a large-scale enzymatic assay terminated at 4 h and subjected to evaporation and lyophilization, respectively. **f** Time-course analysis of the concentration changes of **1**, **6**, **7**, 5′-dA, and SAH from the cyclopropanation process. **g**, **h** are the HR-MS analyses of the intermediate **8** from the reactions using SAM and $CD_3$-SAM, respectively. **i** MS/MS analysis of 8 using SAM. **j** The chemical structure and calculated molecular weight of the intermediate **8**

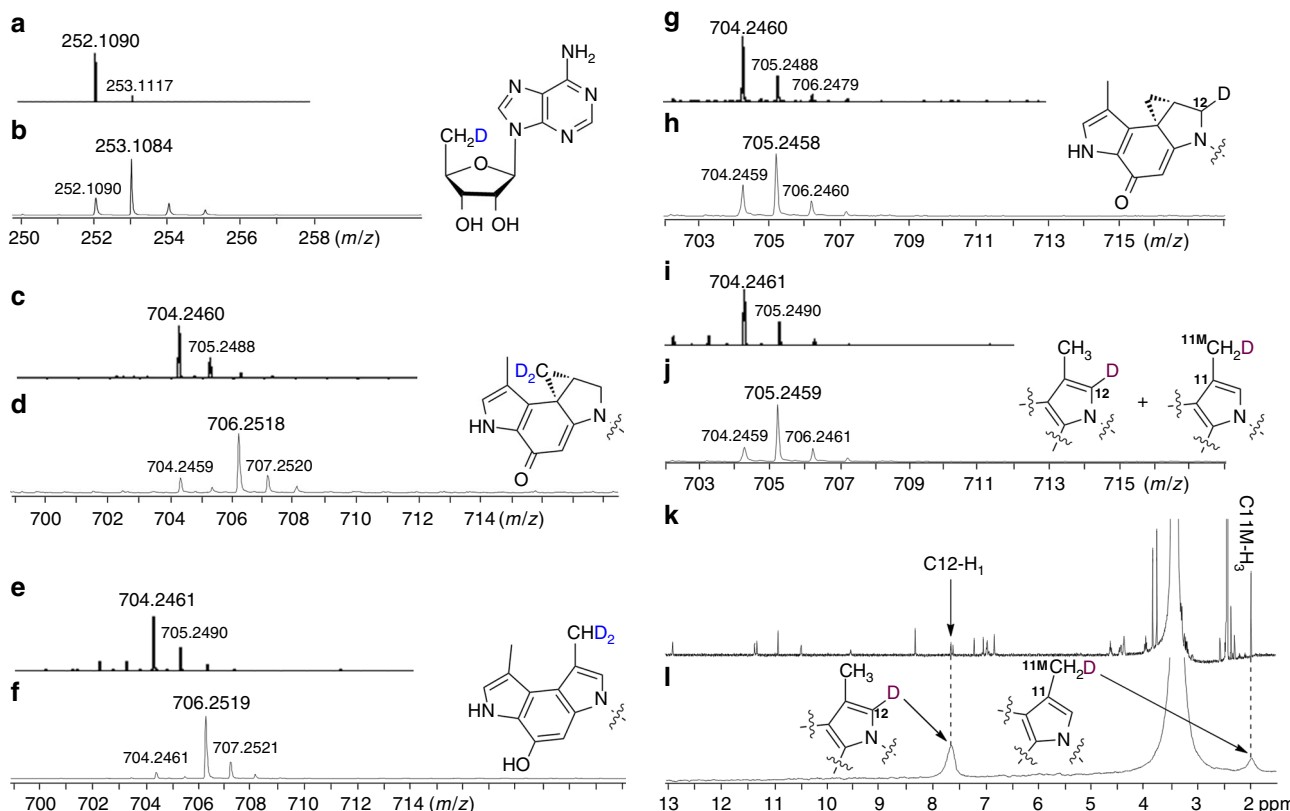

**Fig. 4** Isotope labeling investigations into the cyclopropanation process. **a–f** HR-MS analysis of the products from reactions using SAM or CD$_3$-SAM. **a**, **c**, **e** are the products of 5′-dA, **1**, and **7**, respectively, from the assay using SAM. **b**, **d**, **f** are the products of D-5′-dA, **D$_2$-1**, and **D$_2$-7**, respectively, from the assay using CD$_3$-SAM. **g–j** HR-MS analysis of the products from reactions using H$_2$O or D$_2$O. **g**, **i** are the products of **1** and **7**, respectively, from the assay using H$_2$O. **h**, **j** are the products of **D-1** and **D-7**, respectively, from the assay using D$_2$O. **k** and **l** are the NMR analyses of the enzymatic product **7** from H$_2$O (**7**, $^1$H-NMR) and D$_2$O (**D-7**, $^2$H-NMR)

that it is derived from SAM (Fig. 3h). Both the generated **1** and **7** showed a mass increment of +2 Da ([M + H]$^+$ = 706.2518 for **D$_2$-1** and 706.2519 for **D$_2$-7**) suggesting that only two deuterium atoms from CD$_3$-SAM were incorporated into the products (Fig. 4a–f). In addition, when the enzymatic assays were performed in D$_2$O instead of H$_2$O, **1** and **7** showed a mass shift of +1 Da (Fig. 4g–j); subsequent $^2$H NMR analysis of isolated **D-7** revealed that the chemical shifts of deuterium were identical to those corresponding proton signals of C-12 and the methyl group at C-11 (C-11M) in **7** produced using H$_2$O (Fig. 4k, l). This result suggests that the deuterium atom either located in the C-12 or C-11M position. Collectively, these labeling experiments implicate that the formation of **8** requires the participation of a SAM methylene radical and involves a carbon radical at C-12 (the intermediate **9**) that abstracts a solvent-exchangeable proton, and that the off-pathway compound **7** may be produced from **8** through **10** via isomerization (Fig. 5).

**Site-mutagenesis analysis of the methyltransferase**. The conversion of **8** to **1** requires an intramolecular cyclization reaction with elimination of SAH as a co-product, which resembles an intermolecular methyltransferase activity. Based on our in vivo and in vitro results, this cyclization reaction is probably catalyzed by C10Q. Consistent with this proposal, we did not observe any SAM molecules in the purified C10Q protein (Supplementary Fig. 22). We initially attempted to isolate a little amount of **8**, but it was so unstable that we were not able to obtain sufficient amount for enzymatic assays. Next, we investigated C10Q through mutagenesis experiments. Previous reports have revealed that the plant O-methyltransferases ChOMT and IOMT each

employ a conserved histidine (His) residue to abstract a proton from the methyl acceptor group of substrate[36]. Subsequent multiple sequence alignment identified several conserved motifs from C10Q, such as a variant of the SAM-binding motif DxGxGxG (DxGxNxG for C10Q) and a conserved His residue likely for activation of the methyl acceptor group (Supplementary Fig. 23). As expected, mutation of the SAM-binding motif (C10Q D61A) completely eliminated its activity (Supplementary Fig. 24), while mutation of the conserved His (H138A) only led to production of **7** but not **1** (Fig. 2q). Thus, the C10Q His-138 residue likely serves as a critical catalytic base responsible for deprotonation of the phenolic hydroxyl group at C-6 of the intermediate **8**, and this will promote an intramolecular S$_N$2 reaction to yield **1** with SAH as a leaving group.

Taken together, on the basis of our identification of the intermediate **8** and labeling experiments, we propose a catalytic mechanism for the cyclopropanation process (Fig. 5). Reductive cleavage of the first molecule of SAM$_1$ from the HemN-like radical SAM enzyme yields a highly reactive dAdo radical, which will abstract a hydrogen atom from the activated methyl group of the second molecule of SAM$_2$. A SAM methylene radical is thus produced and then adds to the C-11 position of the substrate **6** to generate a radical intermediate **9**. This radical species abstracts a solvent-exchangeable proton to produce the intermediate **8**. Subsequently, the His-138 residue from C10Q likely deprotonates the phenolic hydroxyl group (C-6) of **8**, which triggers the intramolecular S$_N$2 cyclopropanation to yield **1** with SAH as a co-product. Besides, the intermediate **8** may be non-enzymatically converted to the intermediate **10** containing an exocyclic double bond via elimination of SAH, followed by rapid and

**Fig. 5** Proposed mechanism of the cyclopropanation process. Upon dithionite reduction, the $[4Fe-4S]^{2+}$ from the HemN-like radical SAM enzyme is converted to $[4Fe-4S]^{+}$, which triggers the reductive cleavage of the first molecule of $SAM_1$ to yield a highly reactive dAdo radical. Then, the dAdo radical abstracts a hydrogen atom from the methyl group of the second molecule of $SAM_2$. A SAM methylene radical is thus produced and then adds to the C-11 position of the substrate **6** to generate a radical intermediate **9**. The carbon-centered radical at C-12 in **9** abstracts a solvent-exchangeable proton to produce the intermediate **8**. Subsequently, the His-138 residue from C10Q likely functions as a critical base and deprotonates the phenolic hydroxyl group (C-6) of **8**, which induces the intramolecular $S_N2$ cyclopropanation to yield **1** with elimination of SAH as a co-product. On the other hand, the intermediate **8** may be non-enzymatically converted to the intermediate **10** containing an exocyclic double bond via release of SAH, followed by rapid and thermodynamic driving isomerization to give a methylated off-pathway compound **7**

thermodynamic driving isomerization to give a methylated off-pathway compound **7**.

## Discussion

In this study, we have identified a two-component cyclopropanase system, composed of a HemN-like radical SAM enzyme and a methyltransferase, is able to catalyze a chemically challenging cyclopropanation reaction. Homologues of these two proteins are also encoded by the BGCs of yatakemycin[30] and gilvusmycin[37], as well as several other cryptic BGCs (Supplementary Table 1), indicating this cyclopropanation strategy is conserved in Nature. Previously, HemN-like radical SAM enzymes are categorized into class C radical SAM methyltransferases (RSMTs) owing to their apparent methylation function but with an unclear catalytic mechanism[34,35]. The very recent studies of the class C RSMTs TbtI, NosN, and ChuW have disclosed that they actually catalyze the transfer of a methylene from SAM to substrate[38–42]. Moreover, the HemN-like enzyme Jaw5 identified in the biosynthesis of the polycyclopropanated jawsamycin can be thought of as a variant of C10P (or Swoo_2002), which itself might function as a cyclopropanase[43]. Combining our results, it is much more appropriate to assign these HemN-like proteins methylenetransferases. Radical SAM superfamily proteins utilize a [4Fe-4S] cluster for the reductive cleavage of SAM to generate a highly reactive dAdo radical, which is then harnessed to initiate various radical-mediated biotransformations[33]. Here, our findings support that HemN-like radical SAM enzymes use its first SAM molecule to yield a highly reactive dAdo radical, and then harness this radical to abstract a hydrogen atom from the activated methyl of the second SAM molecule to produce a SAM-based methylene radical. This SAM methylene radical suggests that the methyl carbon atom bonded with sulfonium in SAM can also generate a corresponding carbon-centered radical. The cyclopropanation mechanism in this work is reminiscent of the radical-mediated methylation reactions catalyzed by radical SAM enzymes RlmN and Cfr[34,35]. RlmN catalysis requires a priming

methylation of a conserved Cys-355 residue by SAM to form a methyl thioether group using an $S_N2$ reaction mechanism. Then the dAdo radical produced via reductive cleavage of a second SAM in the same binding site of RlmN is used for the abstraction of a hydrogen from the methylthio group to yield a protein-based thiomethylene radical. This thiomethylene radical adds to the adenosine ring of substrate and therefore affords a critical substrate-protein adduct intermediate, which is further converted to the methylated product with the participation of another key Cys-118 residue. However, the thiomethylene radical as well as the 3-amino-3-carboxypropyl (ACP) radical generated by a non-canonical radical SAM enzyme Dph2 are nucleophilic radicals that attack an electron-deficient C=N double bond to form a C–C bond[44], whereas the SAM methylene radical that is an electrophilic radical due to the electron withdrawing effect of the sulfonium ion favorably adds to an electron-rich C=C double bond (Fig. 5). Analogous to this cyclopropanation reaction, chemical reactions of the charged radical $(CH_3)_2S^+\text{-}CH_2\cdot$ with cyclic alkenes were observed in gas-phase, in which $(CH_3)_2S^+\text{-}CH_2\cdot$ adds to alkenes to form an addition adduct that will lose $CH_3SCH_3$ to complete methylene transfer through C–S bond cleavage[45].

Besides the mechanism we mentioned above for the conversion from **8** to **7** (Fig. 5), an alternative mechanism can also be raised. The carbon-centered radical at C-12 in **9** may trigger the removal of a proton at C-11 by an unknown base with elimination of SAH, and then the formed allylic radical can be quenched at the C-11M position (Supplementary Fig. 25). However, our $^2H$ NMR data have ruled out this possibility because we observed the deuterium atom either located in the C-12 or C-11M position. Furthermore, this alternative mechanism also conflicts with the cyclopropanation process that requires the formation of the detected intermediate **8**. The following deprotonation to eliminate SAH in **8** is very reasonable because analogous reactions in synthetic chemistry have been reported[46,47], e.g., treatment of a brominated compound with DBU (1,8-Diazabicyclo[5.4.0]undec-7-ene) in

acetonitrile to produce a methylated product via dehydro-bromination and isomerization. By analogy, perhaps an unknown base in our enzymatic reaction system induce conversion of the intermediate **8** to **7** via similar mechanism.

Early studies have established two cyclopropanation strategies involving a leaving group, such as displacement of a Cl⁻ leaving group at the $C_\gamma$ position via formation of an enolate intermediate[6–8], and using SAM itself to form a $C_\alpha$ anion that then attacks $C_\gamma$ with release of methylthioadenosine[4,13]. Here, during the formation of the cyclopropyl group in **1** biosynthesis, the methyltransferase C10Q likely abstracts a phenolic hydroxyl proton (C-6) from the SAM addition adduct **8**, and this probably triggers para-cyclopropanation across the conjugated aromatic ring system with SAH as a leaving group. This second step may be considered as an unusual methyl transfer reaction and thus represents a distinct intramolecular nucleophilic displacement mechanism, which is also consistent with its remote evolutionary distance with other methyltransferases (Supplementary Fig. 26).

In summary, we have demonstrated that a two-component cyclopropanase system, composed of a HemN-like radical SAM enzyme and a methyltransferase, catalyzes the formation of the cyclopropyl ring in **1**. This strategy not only illustrates a distinct cyclopropanation process for building a cyclopropane pharmacophore in Nature, but also expands the diversity of SAM-based enzymology and chemistry.

## Methods

**Materials and equipment.** Biochemicals and media were purchased from Sangon Biotech Shanghai Co., Ltd (China), organic solvents for HPLC mobile phase were from Sigma-Aldrich Co., Ltd or Merck Serono Co., Ltd. Chemical reagents such as SAH and CD₃I were from Sigma-Aldrich Co., Ltd and used without further purification. Kits for DNA gel extraction and plasmid DNA extraction were from Shanghai Generay Biotech Co., Ltd. Restriction enzymes were from Thermo Fisher Scientific Co., Ltd. DNA marker and pMD19-T vector were from TaKaRa Biotechnology Co., Ltd. Taq DNA polymerase and Phanta Max Super-Fidelity DNA Polymerase were from Vazyme Biotech Co., Ltd. The CC-1065 producing strain *Streptomyces zelensis* NRRL 11183 was from American Agricultural Research Service (ARS). *Escherichia coli* DH5α competent cells were used for routine subcloning and plasmid preparations, and were grown in LB media with appropriate antibiotics. Plasmids were introduced into *Streptomyces* through intergeneric conjugal transfer using *E. coli* S17-1. Protein expression vectors were purchased from Merck Serono Co., Ltd. Protein expression strains *E. coli* BL21(DE3) and *E. coli* Rosetta(DE3) were also from Merck Serono Co., Ltd. Ni-NTA for protein purification was obtained from Thermo Fisher Scientific Co., Ltd. Enzyme assays were carried out in an anaerobic glove box from Coy Laboratory Product Inc. HPLC analyses were performed in either Agilent 1200 series or Shimazu Prominence LC-20A. Characterization of protein–protein interaction was carried out on Malvern MicroCal ITC200. HR-MS was performed on Bruker (UHR-TOF) maXis 4G or Agilent Q-TOF 6520A.

**Sequence analysis, primer synthesis, and sequencing.** Protein comparison was carried out by BLAST methods (https://blast.ncbi.nlm.nih.gov/Blast.cgi). Multiple sequence alignment was executed using Clustal Omega on the website (http://www.ebi.ac.uk/Tools/msa/clustalo/). Primer synthesis was performed at Sangon Biotech Shanghai Co., Ltd (China) and GENEWIZ Inc.. Sequencing was carried out at GENEWIZ Inc. and Shanghai Invitrogen Biotech Co., Ltd.

**Gene deletion and complementation experiments.** In order to inactivate *c10Q*, a 1.7 kb DNA fragment (amplified using *c10Q*-L-for and *c10Q*-L-rev as primers) digested by *Hin* dIII/*Xho* I and a 1.9 kb DNA fragment (using *c10Q*-R-for and *c10Q*-R-rev as primers) digested by *Xho* I/*Eco* RI were cloned into the thermo-sensitive plasmid pKC1139 (using the *Hin* dIII/*Eco* RI sites) to yield the recombinant plasmid pTG1405. Then, the conjugation was used to introduce pTG1405 from *E. coli* S17-1 into *S. zelensis* NRRL 11183. Apramycin-resistant clones were screened at 30 °C. Candidate exconjugants were then picked to obtain single-crossover mutants in the TSB liquid medium with apramycin at 37 °C. The single-crossover mutants were subsequently grown in the TSB medium without any antibiotics for generations. The anticipated double-crossover mutant *S. zelensis* TG1405 (Δ*c10Q*) was confirmed by PCR analysis (using primers *c10Q*-gt-For and *c10Q*-gt-Rev).

For complementation experiments, in order to introduce *c10Q* into *S. zelensis* TG1405, a 0.9 kb *c10Q*-containing *Eco* RI/*Nde* I DNA fragment (using primers *c10Q*com-For and *c10Q*com-Rev) was cloned into pSET152 to construct the plasmid pTG1406, in which *c10Q* was controlled by the *ermE** promoter. pTG1406

was introduced into *S. zelensis* TG1405 to yield *S. zelensis* TG1406. The fermentation products of TG1406 were subjected to analysis by HPLC for metabolite production. The same procedures were used for the heterologous complementation of *Swoo_2002* (the plasmid pTG1407) into *S. zelensis* TG1402 (the Δ*c10P* mutant) to generate the complemented strain *S. zelensis* TG1407, and for the complementation of *c10P* (the plasmid pTG1408) into *S. zelensis* TG1402 to generate the complemented strain *S. zelensis* TG1408.

**Production and isolation of metabolites.** For fermentation of Δ*c10P* and Δ*c10Q*, spores of *S. zelensis* TG1402 (Δ*c10P*) was inoculated into the TSB media in the 500-mL Erlenmeyer flask for 2-day growth in the conditions of 28 °C and 220 rpm, and the resulting 10 mL of culture was transferred into the fermentation media in the 500-mL flask for 6 days. The components of the fermentation media are as follows[29]: dextrin 30 g, fish meal 10 g, cornmeal 30 g, cottonseed meal 30 g, glucose 10 g, sodium citrate 2.5 g, MgSO₄•7H₂O 1 g, CaCO₃ 5 g, FeSO₄•7H₂O 0.02 g, KCl 0.5 g, CoCl₂•7H₂O 0.02 g, Na₂HPO₄•12H₂O 3 g, and distilled H₂O to make up 1 liter, pH 7.0. The cells of mycelia from culture broth were collected by centrifugation (3220×*g* for 10 min), and the mycelia were then soaked with acetone. The solvent extracts were used for further HPLC analysis. Fermentation of the mutant strain *S. zelensis* TG1405 (Δ*c10Q*) was carried out using the same procedures.

Unlike CC-1065, the intermediate **6** was very unstable during the process of extraction from mycelia with acetone. After many efforts, our procedures for the isolation of the metabolite **6** were summarized. After incubation using fermentation media for 6 days, the culture broth was centrifuged (3220×*g* for 10 min), the culture filtrates were removed, the mycelium pellets were extracted with acetone, the acetone extract was extracted with ethyl acetate within 30 min (**6** is inclined to hydrolyze in basic acetone extract), and then the ethyl acetate phase was dried with granulated anhydrous calcium chloride overnight. The metabolite **6** extracted by ethyl acetate was then subjected to silica gel column chromatography for the first-round isolation (the crude extract was subjected to silica gel column by elution with the mixture of CHCl₃ and MeOH in different ratios, the radio of CHCl₃:MeOH gradient changes as 50:1 → 40:1 → 30:1 → 20:1 → 15:1 → 10:1, the eluent was detected by HPLC and **6** was eluted in 20:1 and 15:1 ratios) and was further purified by semi-preparative HPLC separation (**6** was purified by semi-preparative HPLC separation with reversed-phase column (Venusil XBP C18, 5 μm, 100 Å, 10 × 250 mm) on Shimazu Prominence LC-20A. The column was equilibrated with 85% solvent A (H₂O)/15% solvent B (CH₃CN) for 3 min, the gradient of elution was as follows: 0–3 min: a linear gradient increase from 85% A/15% B to 60% A/40% B, 3–10 min: a linear gradient increase to 45% A/55% B, 10–12 min: a linear gradient increase to 30% A/70% B, 12–21 min: a linear gradient increase to 15% A/85% B, 21–22 min: a linear gradient decrease to 85% A/15% B, 22–23 min: equilibrated with 85% A/15% B. Flow rate was 3 mL•min⁻¹ and detection wavelength was at 374 nm, the HPLC peak at retention time 16.2 min corresponds to **6**). Finally, 6 mg of the metabolite **6** was isolated from 24 L of fermentation broth.

**Construction of protein overexpressing strains.** Taking C10P for example, total DNA extracted from *S. zelensis* NRRL 11183 as template, a DNA fragment (amplified with primers c10P-for and c10P-rev) containing the intact *c10P* gene was first cloned into the T-vector, after verification by sequencing, the 1.37 kb *Eco* RI/*Hin* dIII fragment was recovered from T-vector and then ligated into the same site of pET28a to yield plasmid pTG1409, which was then transformed into *E. coli* Rosetta(DE3) to obtain the expression strain Ro28-P for expressing C10P to give an *N*-terminally 6 × His-tagged protein. Similar procedures were used for the construction of *E. coli* Ro37-Q and *E. coli* RoTB-SW to express C10Q and Swoo_2002, respectively.

For construction of co-expression strain (C10P/C10Q), a 0.7 kb PCR product containing *c10Q* was amplified using the primer pair *c10Q*-gong-for and *c10Q*-gong-rev from the total DNA extracted from *S. zelensis* NRRL 11183 and then cloned into the T-vector, after verification by sequencing, the *Eco* RI/*Hin* dIII DNA fragment was recovered from T-vector and then ligated into the same site of RSFDuet on the MCS I to yield a plasmid *c10Q*RSFDuet. A PCR product containing *c10P* was amplified using the primer pair *c10P*-gong-for and *c10P*-gong-rev from the same template and then cloned into the T-vector, after verification by sequencing, the 1.37 kb *Nde* I/*Xho* I DNA fragment was recovered from T-vector and then ligated into the same site of pET37b to yield a plasmid *c10P*pET37b, and then the 1.37 kb *Nde* I/*Avr* II fragment with 6 × His-tag was recovered from *c10P*pET37b and ligated into the same site of the *c10Q*RSFDuet on the MCS II to yield the plasmid pTG1411, which was then transformed into *E. coli* Rosetta(DE3) to obtain the co-expression strain *E. coli* RoDue-PQ.

**Construction of mutant protein overexpressing strains.** Taking C10Q H138A for example, to replace the His-138 with Ala, the PCR amplification was carried out using the primer pair *c10P*-Q H138A for and *c10P*-Q H138A rev with the plasmid pTG1411 as the template, and the PCR product was subjected to *Dpn* I digestion, then transformed into *E. coli* DH5α. After verification by sequencing, the resulting mutant plasmid pTG1413 was then transformed into *E. coli* Rosetta(DE3) to obtain the expression strain *E. coli* RoDue-PQ_M for expressing C10Q H138A. Similar

procedures were used for the construction of other site-specific mutants of Swoo_2002 and C10Q.

**Protein purification.** *E. coli* Ro28-P (or *E. coli* RoDue-PQ) was used to inoculate into 4 mL of LB media containing 50 μg•mL$^{-1}$ kanamycin, and 25 μg mL$^{-1}$ chloromycetin. The culture was grown at 37 °C overnight, and then transferred to 500 mL of LB media containing the same concentration of antibiotics and 0.25 mM ammonium ferrous sulfate. When the 500 mL culture was grown at 37 °C with an OD at 600 nm of ~0.3, 50 μL of 0.3 M aqueous solution of cysteine and 50 μL of 0.25 M ammonium ferrous sulfate were added, and then the culture was further grown at 37 °C until the OD at 600 nm reached ~ 0.6, the protein expression was then induced upon addition of 30 μL of 1 M IPTG followed by incubation at 16 °C for another 24 h. Similar procedures were applied for the expression of Swoo_2002 using the strain *E. coli* RoTB-SW, the only difference was that incubation was performed at 22 °C for 18 h.

After induction upon addition of IPTG and incubation for protein expression, the cells were harvested from the 500 mL of the culture broth (3220×*g* for 10 min). The purification was then performed under 4 °C in an anaerobic glove box with less than 1 ppm of O$_2$. The collected cells were resuspended in 30 mL of the lysis buffer (HEPES 50 mM, KCl 300 mM, imidazole 2 mM, mercaptoethanol 10 mM, glycerol 5%, pH 7.5) containing 1 mg mL$^{-1}$ of lysozyme, and disrupted by ultra-sonication. The bacterial suspension was sealed in the anaerobic glove box and centrifuged at 7250×*g* for 60 min and then taken back into the anaerobic glove box, the supernatant was incubated with 4 mL of Ni-NTA resin pre-equilibrated with the lysis buffer at 4 °C for 1 h, and then subjected to affinity purification on a column by elution with different concentrations of imidazole (50 mM 12 mL, 100 mM 8 mL, 150 mM 4 mL, and 400 mM 3 mL) prepared by mixing lysis buffer with elution buffer (HEPES 50 mM, KCl 300 mM, imidazole 400 mM, mercaptoethanol 10 mM, glycerol 20%, pH 7.5). The protein was finally eluted by 3 mL 400 mM elution buffer and then subjected to a desalination column to obtain the protein in 3 mL Tris•HCl buffer (Tris 50 mM, NaCl 100 mM, glycerol 10%, pH 8.0). The purity of the purified protein was determined by 12% SDS-PAGE analysis. However, it turned out that the protein purified under aerobic conditions and then reconstituted under anaerobic conditions possessed the same activity as those purified under anaerobic conditions and then reconstituted.

**Reconstitution of the radical SAM proteins.** The radical SAM enzymes (C10P and Swoo_2002) need to be reconstituted under anaerobic conditions before enzymatic assays and the reconstitution was performed at 4 °C. The protein in 3 mL of Tris•HCl buffer was added 30 μL of 1 M DTT to 10 mM, 15 min later, 30 μL of 50 mM ammonium ferrous sulfate was added to the final concentration of 0.5 mM, after 45 min, 10 μL of 50 mM sodium sulfide was added every 30 min for three times to the final concentration of 0.5 mM, and then reconstituted for at least 3 h. Afterwards, the reconstituted mixture was subjected to a desalination column to obtain the dark brown protein in 3 mL Tris•HCl buffer.

**UV–visible analysis of the radical SAM enzyme.** UV–visible analysis of the reconstituted radical SAM enzyme (and the reconstituted radical SAM enzyme reduced by sodium dithionite) was conducted on Thermo NanoDrop 2000 spectrophotometer.

**Iron assay for Swoo_2002.** The quantity of iron in the reconstituted Swoo_2002 was measured by standard curve method[48]. After preparation of solution A (equivalent volume mixture of 1.2 M HCl and 0.285 M KMnO$_4$), solution B (9.7 g of ammonium acetate and 8.8 g of ascorbic acid were dissolved in 10 mL of H$_2$O, and then 80 mg of ferrozine and 80 mg of neocuproine were added followed by diluting with water to 25 mL) and 6 μg mL$^{-1}$ ammonium ferrous sulfate as standard solution, 1 mL of standard solution diluted into different concentrations were added 0.5 mL of solution A and incubated at 60 °C for 2 h, and then the mixture was added 0.1 mL of solution B and incubated at room temperature for more than 30 min, the absorption value at 562 nm was measured on JASCO V530 and a calibration curve was made. Then 1 mL of sample of Swoo_2002 at 20 μM was subjected to the same procedure as standard solution and measured the absorption value at 562 nm, and then the quantity of iron in Swoo_2002 was calculated based on the calibration curve.

**Sulfur assay for Swoo_2002.** The quantity of sulfur in the reconstituted Swoo_2002 was measured by a standard curve method as well[49]. Six microgram per milliliter of 6 μg mL$^{-1}$ sodium sulfide was prepared as standard solution, 200 μL of standard solution diluted into different concentrations were added 600 μL of 1% zinc acetate and 50 μL of 7% NaOH and then incubated at room temperature for 15 min, and then 150 μL of 0.1% DMPD (*N*, *N*-dimethyl-*p*-phenylenediamine monohydrochloride) in 5 M HCl and 150 μL of 10 mM FeCl$_3$ solution in 1 M HCl were added. After 20 min of incubation, the absorption value at 670 nm was measured and a calibration curve was made. Then 200 μL of sample of Swoo_2002 at 20 μM was subjected to the same procedure as standard solution and measured the absorption value at 670 nm, and then the quantity of sulfur in Swoo_2002 was calculated based on the calibration curve.

**Isothermal titration calorimetry analysis.** Proteins were exchanged from Tris•HCl buffer to PBS buffer (10 × PBS buffer: 80 g of NaCl, 2 g of KCl, 29 g of Na$_2$HPO$_4$•12H$_2$O, 2 g of KH$_2$PO$_4$ in 1 L H$_2$O, pH 7.4) by subjected to a desalination column for three times (to make sure that the protein sample excludes glycerol and contains less than 5‰ DMSO). The ITC analysis was performed on Malvern MicroCal ITC200 with 600 μM of C10Q titrating 200 μL of 60 μM Swoo_2002 and the results worked out automatically in the process of titration.

**In vitro enzymatic assays.** All in vitro enzyme activity assays were performed in an anaerobic glove box with less than 1 ppm of O$_2$. Enzymatic reactions were conducted in Tris•HCl buffer (Tris 50 mM, NaCl 100 mM, glycerol 10%, pH 8.0) with the following components: 1 μM reconstituted Swoo_2002, 2 μM C10Q, 10 μM substrate, 1 mM SAM, 5 mM DTT, 5 mM MgCl$_2$, 5 mM Na$_2$S$_2$O$_4$ and 7% DMSO. The reactions were incubated at 28 °C for 12 h.

**HPLC analysis of enzymatic products.** HPLC analysis of CC-1065 and **7** were performed by reversed-phase column (Grace Alltima, C18, 5 μm, 100 Å, 10 × 250 mm) on Agilent 1200 series. The column was equilibrated with 85% solvent A (H$_2$O)/15% solvent B (CH$_3$CN) for 3 min. The gradient of elution was as follow: 0–3 min: a linear gradient increase from 85% A/15% B to 60% A/40% B, 3–10 min: a linear gradient increase to 45% A/55% B, 10–12 min: a linear gradient increase to 30% A/70% B, 12–21 min: a linear gradient increase to 15% A/85% B, 21–22 min: a linear gradient decrease to 85% A/15% B, 22–25 min: 85% A/15% B. Flow rate was 1 mL•min$^{-1}$ and detection wavelength was at 374 nm, the HPLC peak at retention time 13.7 min and 14.8 min correspond to CC-1065 and **7**, respectively.

HPLC analysis of SAH and 5′-dA was performed by reversed-phase column (Diamonsil, C18, 5 μm, 4.6 × 250 mm) on Agilent 1200 series. The column was equilibrated with 95% solvent A (H$_2$O + 1‰ TFA)/5% solvent B (CH$_3$CN + 1‰ TFA) for 3 min. The gradient of elution was as follow: 0–5 min: 95% A/5% B, 5–20 min: a linear gradient increase to 80% A/20% B, 20–24 min: a linear gradient increase to 10% A/90% B, 24–27 min: 10% A/90% B, 27–29 min: a linear gradient decrease to 95% A/5% B, 29–30 min: 95% A/5% B. Flow rate was 1 mL•min$^{-1}$ and detection wavelength was at 260 nm, the HPLC peak at retention time 9.0 min and 12.2 min correspond to SAH and 5′-dA, respectively.

**Isolation and characterization of enzymatic products.** Enzymatically produced SAH, 5′-dA and CC-1065 were purified by semi-preparative HPLC separation and HR-MS analyses were conducted on Bruker (UHR-TOF) maXis 4 G, it turned out that the product SAH presented exactly the same molecular mass as the standard SAH (calcd. for C$_{14}$H$_{21}$N$_6$O$_5$S$^+$ 385.1294 [M + H]$^+$, found 385.1293; calcd. for C$_{14}$H$_{20}$N$_6$NaO$_5$S$^+$ 407.1114 [M + Na]$^+$, found 407.1112), the product 5′-dA presented exactly the same molecular mass as the standard 5′-dA (calcd. for C$_{10}$H$_{14}$N$_5$O$_3^+$ 252.1097 [M + H]$^+$, found 252.1090; calcd. for C$_{10}$H$_{13}$N$_5$NaO$_3^+$ 274.0916 [M + Na]$^+$, found 274.0908), and the product CC-1065 also presented exactly the same molecular mass as the standard CC-1065 (calcd. for C$_{37}$H$_{34}$N$_7$O$_8^+$ 704.2469 [M + H]$^+$, found 704.2460; calcd. for C$_{37}$H$_{33}$N$_7$ NaO$_8^+$ 726.2288 [M + Na]$^+$, found 726.2267) and the MS/MS result of the product CC-1065 was consistent with the standard CC-1065 as well.

Enzymatically produced **7** was purified by semi-preparative HPLC separation with reversed-phase column (Venusil XBP C18, 5 μm, 100 Å, 10 × 250 mm) on Shimazu Prominence LC-20A. The column was equilibrated with 60% solvent A (H$_2$O)/40% solvent B (CH$_3$CN) for 3 min, the gradient of elution was as follows: 0–3 min: a linear gradient increase from 60% A /40% B to 50% A/50% B, 3–6 min: a linear gradient increase to 30% A/70% B, 6–10 min: a linear gradient increase to 15% A/85% B, 10–12 min: a linear gradient increase to 10% A/90% B, 12–14 min: a linear gradient increase to 5% A/95% B, 14–16 min: a linear gradient decrease to 60% A/40% B. Flow rate was 3 mL min$^{-1}$ and detection wavelength was at 374 nm. HR-MS analysis of the purified product **7** showed that it presented exactly the same molecular mass as the standard CC-1065 (C$_{37}$H$_{34}$N$_7$O$_8^+$ 704.2460), but the further MS/MS data confirmed that the product **7** is the methylated product of substrate **6** on C-11. In the $^1$H NMR (DMSO-$d_6$) data of substrate **6**[29], δ 2.46 (s, 3 H), 3.22 (t, 2 H, *J* = 8.86 Hz), 3.33 (t, 2 H, *J* = 7.17 Hz), 3.82 (s, 3 H), 3.91 (s, 3 H), 4.03 (t, 2 H, *J* = 8.86 Hz), 4.70 (t, 2 H, *J* = 7.17 Hz), 6.90 (s, 2 H), 6.96 (d, 1 H, *J* = 2.28 Hz), 7.05 (s, 1 H), 7.08 (s, 1 H), 7.11 (s, 1 H), 7.71 (s, 1 H), 7.77 (d, 1 H, *J* = 2.28 Hz), 8.42 (s, 1 H, OH), 10.81 (s, 1 H), 11.14 (s, 1 H, OH), 11.34 (s, 1 H, OH), 11.95 (s, 1 H), 12.93 (s, 1 H, OH), the signal 6.96 (d, 1 H, *J* = 2.28 Hz) corresponds to the 11-CH of **6** coupling with the adjacent 12-CH. However, in the $^1$H NMR (DMSO-$d_6$) data of the product **7**, the signal of 11-CH is missing while the signal of 12-CH remains. Besides, there is an extra signal that appeared at 2.05 ppm corresponding to the 11-CH$_3$ connecting with C = C (11-C). In addition, the H-H COSY data of substrate **6** showed an obvious signal of H-H coupling between 11-CH and 12-CH whereas the H-H coupling signal is missing in the H-H COSY data of the product **7**. In conclusion, all data enumerated above indicate that the product **7** is the methylated product of substrate **6** on 11-C.

**Synthesis of CD$_3$-SAM.** The reaction was conducted under dark condition[50], 5.25 mg of SAH (1 eq) was dissolved in 0.5 mL of HCOOH and 0.5 mL of CH$_3$COOH at 0 °C, and then 41.5 μL of CD$_3$I (50 eq) was added, after this 5.4 mg of AgClO$_4$ (2 eq) was added and the mixture was stirred at 0 °C for 5 min before

recovery to room temperature. After stirring at room temperature for 24 h, the reaction was quenched by 3 mL of $H_2O$ and extracted by diethyl ether (3 × 5 mL), and then the supernatant of aqueous phase after centrifugation was lyophilized to remove HCOOH, $CH_3COOH$ and $H_2O$, the remaining solid containing $CD_3$-SAM was subjected to semi-preparative HPLC separation with reversed-phase column (Venusil XBP C18, 5 μm, 100 Å, 10 × 250 mm) on Shimazu Prominence LC-20A. The column was equilibrated with 95% solvent A ($H_2O$ + 1‰ TFA)/5% solvent B ($CH_3CN$ + 1‰ TFA) for 3 min, the gradient of elution was as follows: 0–1 min: 95% A/5% B, 1–12 min: a linear gradient increase to 92.38% A/7.62% B, 12–14 min: a linear gradient increase to 30% A/70% B, 14–15 min: a linear gradient decrease to 95% A/5% B. Flow rate was of 3 mL•min$^{-1}$ and detection wavelength was 260 nm. The $CD_3$-SAM was collected at 5.7 min. HR-MS of $C_{15}H_{20}D_3N_6O_5S^+$ ([M]$^+$), calculated 402.1633, found 402.1633.

**Labeling experiments**. For $CD_3$-SAM related experiments, $CD_3$-SAM was used instead of regular SAM during in vitro enzyme activity assays, the products CC-1065, 7, SAH, and 5′-dA were purified by HPLC and subjected to HR-MS, it turned out that the majority of the product CC-1065 showed a mass shift of +2$m/z$ ($C_{37}H_{32}D_2N_7O_8^+$ 706.2518) while the minority remained unchanged, the majority of the product 7 showed a mass shift of +2$m/z$ ($C_{37}H_{32}D_2N_7O_8^+$ 706.2519) while the minority unchanged, the majority of the product 5′-dA showed a mass shift of +1$m/z$ ($C_{10}H_{13}DN_5O_3^+$ 253.1084) and the minority unchanged while the molecular mass of the product SAH remained unchanged ($C_{14}H_{21}N_6O_5S^+$ 385.1293). Observing these small amount of unlabeled products is in good agreement with the fact that C10P and Swoo_2002 can co-purified with a small amount of unlabeled SAM showed by HPLC analysis.

For $D_2O$ experiments, all chemical reagents for enzyme assays were dissolved in $D_2O$ and proteins were exchanged into Tris•HCl buffer prepared in $D_2O$. The products CC-1065, 7, SAH, and 5′-dA were purified by HPLC and subjected to HR-MS, it turned out that the majority of the product CC-1065 showed a mass shift of +1$m/z$ ($C_{37}H_{33}DN_7O_8^+$ 705.2458) while the minority stayed unchanged, the majority of the product 7 showed a mass shift of +1$m/z$ ($C_{37}H_{33}DN_7O_8^+$ 705.2459) while the minority unchanged, the molecular mass of the product 5′-dA and SAH remained the same ($C_{10}H_{14}N_5O_3^+$ 252.1090 for 5′-dA and $C_{14}H_{21}N_6O_5S^+$ 385.1293 for SAH).

**Identification of a SAM-substrate intermediate**. When the enzyme assay was terminated in advance and subjected to HR-MS, we searched for a molecular mass signal consistent with the putative SAM-substrate intermediate ($C_{51}H_{54}N_{13}O_{13}S^+$ 1088.3761). This peak in HPLC was previously ignored for its quite low quantity and instability, and this intermediate is so unstable that it will decompose when the enzyme assay mixture was subjected to rotary evaporation. We then enriched this probable SAM intermediate by HPLC from large-scale enzyme assays and confirmed its identity through MS/MS. Subsequently, the $CD_3$-SAM experiments were conducted in large scale to enrich the probable deuterated SAM intermediate likewise, then the intermediate was purified by HPLC and subjected to HR-MS, it turned out that the majority of the deuterated SAM intermediate showed a mass shift of +2$m/z$ ($C_{51}H_{52}D_2N_{13}O_{13}S^+$ 1090.3809) while the minority stayed unchanged.

**Data availability**. The data that support the findings of this study are available from the corresponding author upon request.

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

## Acknowledgements
This work was supported in part by grants from National Natural Science Foundation of China (21632007, 21502217 and 21621002), Science and Technology Commission of Shanghai (15ZR1449400 and 15JC1400400), Chinese Academy of Sciences (XDB20000000, QYZDJ-SSW-SLH037 and K.C. Wong Education Foundation), and State Key Laboratory of Microbial Metabolism, Shanghai Jiao Tong University (MMLKF15-02 and MMLKF15-11). We dedicated this paper to Prof. Hai-Bao Chen on the occasion of his 80th birthday. We thank Prof. Ang Li and Prof. Chao-Zhong Li for their helpful suggestions. We also thank Yi-Ning Liu at Institute of Plant Physiology and Ecology, Shanghai Institutes for Biological Sciences, Chinese Academy of Sciences for technical assistance in MS data collection.

## Author contributions
G.-L.T. and S.W. conceived the study and designed the experiments. W.-B.J., S.W., H.Y. and X.-H.J. participated in the genetic experiments. W.-B.J. characterized the compounds and performed the biochemical assays. W.-B.J., S.W., H.Y. and G.-L.T. interpreted the data and wrote the manuscript.

## Additional information

**Competing interests:** The authors declare no competing interests.

