## [Peer Review File · Nature Communications]

Reviewers' comments:

Reviewer #1 (Remarks to the Author):

This submission from Tang and co-workers details the biochemical characterization of two enzymes from the biosynthetic pathway that constructs the antitumor natural product CC-1065 that are involved in the biosynthesis of a cyclopropane ring. This structural feature is critical for the bioactivity of these natural products. These authors had previously used genetics to identify two genes from this pathway (c10P and C10Q) that were linked to cyclopropane formation in vivo. They then characterized the predicted radical SAM enzyme C10P (and homolog SW_2002) and O-methyltransferase C10Q in vitro. They find that both enzymes are needed to observe the formation of small amounts of the cyclopropane-containing natural product CC-1065 (1) and a proposed shunt product 4. They also identify small amounts of a potential intermediate 5, which appears to be the product of addition of the methyl group of SAM to the substrate (3). Based on the characterization of this species and the outcome of labeling experiments, the authors propose a mechanism for cyclopropane formation that involves C10P-catalyzed addition of a SAM-derived radical into substrate 3 to generate intermediate 5, followed by intramolecular alkylation to form the cyclopropane mediated by C10Q. They further discover that homologs of these two genes are distributed across multiple biosynthetic gene clusters, suggesting that this strategy may operate within additional metabolic pathways.

Overall, the authors have uncovered a chemically interesting new biosynthetic strategy for cyclopropane formation. These findings will appeal to a broad audience of natural product chemists and enzymologists, I would support accepting this paper to Nature Communications after suitable revisions are made. Notably, there are a few aspects of this work that appear to be either preliminary or highly unusual, and I outline the points that the authors should address below.

Do the authors have an explanation for why the yields of 1 are so poor in comparison to those of the proposed 'off pathway' product 4? The explanation provided is unsatisfying as the authors indicate they do not detect 4 in vivo. This suggests that they may be working with a system that is poorly behaved, which is problematic because the observed activity may not be relevant to native catalysis. Could the authors be missing an important assay component or an additional protein? Even if this issue can't be resolved, the authors should clearly state the limits or caveats to the conclusions they are drawing from their data.

Can the authors observe the accumulation and consumption of the proposed 'intermediate' 5 in an assay time course? Detecting this species in very low abundance at a single 4 h time point seems like not enough evidence to support the proposed role as an intermediate. What about earlier time points? Even if 5 is not stable enough to isolate and resubject to the assay conditions, a time course would be useful data to include. With a more detailed experiment, the evidence that 5 is on pathway to product is very limited.

The specific mutations used to disrupt SAM binding in C10Q should be specified in the main text and the methods used to construct these mutants should be specified in the Supplementary Information.

In supplementary Figure 16, one cannot see the conserved residues in the multiple sequence alignment (color blocks are too opaque). The authors should fix this.

The authors should include a table of the C10P/C10Q homologs identified in biosynthetic gene clusters in the SI, along with the appropriate accession #s.

Finally, this manuscript will need to undergo thorough editing to improve grammar and sentence

structure (including changing the title). The authors should also use appropriate chemical language when describing reaction mechanisms (for example, use 'abstracts or removes' rather than 'grabs' to describe hydrogen atom abstraction chemistry).

Reviewer #2 (Remarks to the Author):

The authors characterize a two-enzyme system that utilizes radical SAM chemistry to catalyze a cyclopropanation reaction in biosynthesis of the antibiotic CC-1065. Using a combination of mutation/complementation and in vitro analysis, the authors demonstrate that the reaction proceeds by a 2-SAM mechanism, in which the first SAM is used to produce the 5'-deoxyadenosyl radical intermediate. This radical intermediate abstracts a hydrogen atom from the methyl of a second SAM, and this SAM radical then couples to substrate. Subsequent steps lead to the release of S-adenosylhomocysteine and the cyclopropane-containing product. This is an interesting and novel reaction for radical SAM enzymes, and will be of significant interest to the field. The authors have extensively documented their experimental results both in the main body of the paper and in the SI. There are a few issues of language and lack of clarity in the writing, however overall this is a very interesting story. The observation of the intermediate 5 is particularly intriguing.

Following are some items that should be addressed by the authors:

The first paragraph of the introduction is not very clear. The authors outline two classes of cyclopropane biosynthesis, one that involves only a single substrate and one that uses SAM as a co-substrate. The subfamilies of the former are delineated, and one of them involves SAM as a co-substrate, so it seems that the delineation of these classes and subfamilies is not very clear. In the second to last sentence, when they refer to "The later, only known by..." (should be 'The latter...'), it is not clear to what they are referring. The last sentence of the Introduction, beginning "Here we describe..." is not clearly written.

The Discussion section is particularly problematic in terms of clarity; careful revision of this with the help of a native English speaker would be helpful.

Figure 1b: Why is there no standard for compound 2?

Figure 4: The legend should provide more detail so that the reader can readily discern what each of the 10 mass spectra are illustrating.

Figure 5: A figure legend explaining what is illustrated is needed here.

Reviewer #3 (Remarks to the Author):

The cyclopropyl group of the antitumor agent CC1065 is directly involved in DNA alkylation and its biosynthesis has intrigued biosynthetic chemists and mechanistic enzymologists since this compound was discovered several decades ago. The paper by Professor Tang and his colleagues, identifying the two genes involved in cyclopropane formation, describing a set of mechanistic experiments and proposing an elegant mechanism consistent with all available data is therefore of high interest and suitable for publication in Nature Communications.

Page 3, Paragraph 1: A figure describing known strategies for the biosynthesis of cyclopropanes in various natural products would be of value here.

Page 3, Paragraph 2, line 4 and many other places in the text: As far as I can tell, the cyclopropyl group in CC-1065 is not a spirocyclopropane. For example, the normal use of “spiro” is shown below.

Figure 1 (and others). All compounds should be numbered sequentially and panel labeling should be consistent (i.e. I is after the label and III is before the label in Figure 1).

Page 7, Paragraph 1 and Figures 4b and 5: A 1,3 hydride shift is geometrically unfavorable. The deuterium distribution in 4 between C12 and C11m therefore needs additional explanation. One possibility might be a non-pericyclic reaction involving protonation at C11 of 7, hydride migration from C12 to the cation at C11m followed by deprotonation at C11. The authors refer to an NMR spectrum supporting the labeling of 4 shown in Figure 4b. I cannot find this spectrum. What is the ratio of H vs D migration?

Page 7, Paragraph 1, line 6: The assignment of H138 as the base involved in phenol deprotonation is tentative. I suggest replacing “deprotonates the phenolic” with “is likely to deprotonate the phenolic”.

Page 7, Paragraph 1, line 5 (and several other places): I suggest replacing “grabs” with “abstracts”.

Page 8, Paragraph 2, line 5: replace “double” with “double bond”

Page 8, Paragraph 2 line 9 from the end: I suggest replacing “which requires the phenolic hydroxyl and C-S bond about to break arranged collinearly on intermediate 5.” With “which requires that the HOMO of the phenol can overlap with the C-S σ^* orbital in a strained transition state”.

Page 8, Paragraph 2 line 8 from the end: The authors state: “---whereas in the generation of off-pathway product **4**, a more convenient E2 elimination promoted by a general base may occur much faster under the biochemical assay conditions *in vitro*.” This is not at all clear to me. In order for the E₂ elimination to occur, the C11H and C-S bonds must be trans and coplanar. This requires a major movement of most of the atoms of SAM in the active site. This merits some discussion.

Page 10, Line 4: replace “centrifugalized” with “centrifuged”

Page 12, line 3: Please state that HPLC conditions are described in SI.

Page S6: Please provide chromatography conditions for the purification of 3

Page S10, line 4: Define DMPD.

Page S30: Labels for the gene clusters are distorted. I recommend listing all 12 identified clusters.

Reviewers' comments:

Reviewer #1 (Remarks to the Author):

This submission from Tang and co-workers details the biochemical characterization of two enzymes from the biosynthetic pathway that constructs the antitumor natural product CC-1065 that are involved in the biosynthesis of a cyclopropane ring. This structural feature is critical for the bioactivity of these natural products. These authors had previously used genetics to identify two genes from this pathway (c10P and C10Q) that were linked to cyclopropane formation in vivo. They then characterized the predicted radical SAM enzyme C10P (and homolog SW_2002) and O-methyltransferase C10Q in vitro. They find that both enzymes are needed to observe the formation of small amounts of the cyclopropane-containing natural product CC-1065 (1) and a proposed shunt product 4. They also identify small amounts of a potential intermediate 5, which appears to be the product of addition of the methyl group of SAM to the substrate (3). Based on the characterization of this species and the outcome of labeling experiments, the authors propose a mechanism for cyclopropane formation that involves C10P-catalyzed addition of a SAM-derived radical into substrate 3 to generate intermediate 5, followed by intramolecular alkylation to form the cyclopropane mediated by C10Q. They further discover that homologs of these two genes are distributed across multiple biosynthetic gene clusters, suggesting that this strategy may operate within additional metabolic pathways.

Overall, the authors have uncovered a chemically interesting new biosynthetic strategy for cyclopropane formation. These findings will appeal to a broad audience of natural product chemists and enzymologists, I would support accepting this paper to Nature Communications after suitable revisions are made. Notably, there are a few aspects of this work that appear to be either preliminary or highly unusual, and I outline the points that the authors should address below.

Do the authors have an explanation for why the yields of 1 are so poor in comparison to those of the proposed 'off pathway' product 4 ? The explanation provided is unsatisfying as the authors indicate they do not detect 4 in vivo. This suggests that they may be working with a system that is

poorly behaved, which is problematic because the observed activity may not be relevant to native catalysis. Could the authors be missing an important assay component or an additional protein? Even if this issue can't be resolved, the authors should clearly state the limits or caveats to the conclusions they are drawing from their data.

Response: It is frequently considered as a difficult problem to deal with radical SAM superfamily proteins because they require additional protein components (as electron donors) to perform their catalysis *in vivo*. However, scientists usually use alternative electron donors (such as chemical reductants methyl viologen or dithionite, or an *E. coli* reduction system composed of flavodoxin, flavodoxin reductase, and NADPH) instead of native protein electron donors. Therefore, the enzymatic production of desired products may be poor (e.g., Bridwell-Rabb, J., *et al.*, A B₁₂-dependent radical SAM enzyme involved in oxetanocin A biosynthesis. *Nature*, 2017, 544: 322-326.), and sometimes off-pathway products are obviously generated (e.g., Zhang, Q., *et al.*, Radical-mediated enzymatic carbon chain fragmentation-recombination. *Nat Chem Biol*, 2011, 7: 154-160.).

Probably, the generation of off-pathway products is triggered by the inherently high reactivity of the biosynthetic radical intermediate produced when non-native electron donors collaborate badly with the radical SAM superfamily proteins. In our enzymatic cyclopropanation system, the interactions are more complex, at least involving interactions between the electron donor and C10P, and between C10P and C10Q, which may lead to high yield of this off-pathway product.

Comparison of shared genes from the biosynthetic gene clusters of CC-1065 and yatakemycin, and several related cryptic clusters revealed that the radical SAM enzyme and methyltransferase are probably the only proteins involved in the cyclopropyl formation. This is also consistent with our systematic gene deletion experiment results. These data thus suggest that the native protein electron donors are encoded outside of these gene clusters.

Can the authors observe the accumulation and consumption of the proposed 'intermediate' 5 in an assay time course? Detecting this species in very low abundance at a single 4 h time point seems like not enough evidence to support the proposed role as an intermediate. What about earlier time points? Even if 5 is not stable enough to isolate and resubject to the assay conditions, a time course would be useful data to include. With a more detailed experiment, the evidence that 5 is on pathway to product is very limited.

Response: We agree with the reviewer. We have performed a time-course analysis of the production of this critical intermediate **8** (reassigned in this revised submission), and found that the concentration of **8** first increased (from 0.5 to 6 hours) and then decreased (from 6 to 12 hours) as the reaction proceeded. We have provided a figure as Supplementary Figure 20 in this submission. Therefore, combining our MS/MS and labeling experiment results and the requirement of a methyltransferase (C10Q), we believe that it is reasonable to propose that **8** is an enzyme-catalyzed intermediate.

The specific mutations used to disrupt SAM binding in C10Q should be specified in the main text and the methods used to construct these mutants should be specified in the Supplementary

Information.

Response: We agree with the reviewer. We have included the related information in this submission.

In supplementary Figure 16, one cannot see the conserved residues in the multiple sequence alignment (color blocks are too opaque). The authors should fix this.

Response: We agree with the reviewer. We have changed the colors (Supplementary Fig. 23 in this submission).

The authors should include a table of the C10P/C10Q homologs identified in biosynthetic gene clusters in the SI, along with the appropriate accession #s.

Response: We agree with the reviewer. We have included the related information in this submission (Supplementary Fig. 25).

Finally, this manuscript will need to undergo thorough editing to improve grammar and sentence structure (including changing the title). The authors should also use appropriate chemical language when describing reaction mechanisms (for example, use 'abstracts or removes' rather than 'grabs' to describe hydrogen atom abstraction chemistry).

Response: We agree with the reviewer. We have thoroughly improved the manuscript in this submission.

Reviewer #2 (Remarks to the Author):

The authors characterize a two-enzyme system that utilizes radical SAM chemistry to catalyze a cyclopropanation reaction in biosynthesis of the antibiotic CC-1065. Using a combination of mutation/complementation and *in vitro* analysis, the authors demonstrate that the reaction proceeds by a 2-SAM mechanism, in which the first SAM is used to produce the 5'-deoxyadenosyl radical intermediate. This radical intermediate abstracts a hydrogen atom from the methyl of a second SAM, and this SAM radical then couples to substrate. Subsequent steps lead to the release of S-adenosylhomocysteine and the cyclopropane-containing product. This is an interesting and novel reaction for radical SAM enzymes, and will be of significant interest to the field. The authors have extensively documented their experimental results both in the main body of the paper and in the SI. There are a few issues of language and lack of clarity in the writing, however overall this is a very interesting story. The observation of the intermediate 5 is particularly intriguing.

Following are some items that should be addressed by the authors:

The first paragraph of the introduction is not very clear. The authors outline two classes of cyclopropane biosynthesis, one that involves only a single substrate and one that uses SAM as a co-substrate. The subfamilies of the former are delineated, and one of them involves SAM as a

co-substrate, so it seems that the delineation of these classes and subfamilies is not very clear. In the second to last sentence, when they refer to "The later, only known by..." (should be "The latter..."), it is not clear to what they are referring.

Response: We agree with the reviewer. We have thoroughly improved the manuscript in this submission. We have divided enzymatic cyclopropanation strategies into three classes according to the degree of dependence on SAM, and included a Supplementary Figure 1 to help readers to catch our descriptions.

The last sentence of the Introduction, beginning "Here we describe..." is not clearly written. The Discussion section is particularly problematic in terms of clarity; careful revision of this with the help of a native English speaker would be helpful.

Response: We agree with the reviewer. We have improved the manuscript (including Introduction and Discussion) in this submission.

Figure 1b: Why is there no standard for compound 2?

Response: Because this compound is unrelated to our present work, we have omitted it in this improved submission, and supplied a new figure (Figure 2a in this submission).

Figure 4: The legend should provide more detail so that the reader can readily discern what each of the 10 mass spectra are illustrating. Figure 5: A figure legend explaining what is illustrated is needed here.

Response: We agree with the reviewer. We have provided related legends for all figures in this submission.

Reviewer #3 (Remarks to the Author):

The cyclopropyl group of the antitumor agent CC1065 is directly involved in DNA alkylation and its biosynthesis has intrigued biosynthetic chemists and mechanistic enzymologists since this compound was discovered several decades ago. The paper by Professor Tang and his colleagues, identifying the two genes involved in cyclopropane formation, describing a set of mechanistic experiments and proposing an elegant mechanism consistent with all available data is therefore of high interest and suitable for publication in Nature Communications.

Page 3, Paragraph 1: A figure describing known strategies for the biosynthesis of cyclopropanes in various natural products would be of value here.

Response: We agree with the reviewer. We have improved the manuscript in this submission. We have divided enzymatic cyclopropanation strategies into three classes according to the degree of dependence on SAM, and included a Supplementary Figure 1 to help readers to catch our descriptions.

Page 3, Paragraph 2, line 4 and many other places in the text: As far as I can tell, the cyclopropyl group in CC-1065 is not a spirocyclopropane. For example, the normal use of “spiro” is shown below.

Response: We agree with the reviewer. We have changed them to cyclopropane or cyclopropyl.

Figure 1 (and others). All compounds should be numbered sequentially and panel labeling should be consistent (i.e. I is after the label and III is before the label in Figure 1).

Response: We agree with the reviewer. We have numbered all the compounds sequentially.

Page 7, Paragraph 1 and Figures 4b and 5: A 1,3 hydride shift is geometrically unfavorable. The deuterium distribution in 4 between C12 and C11m therefore needs additional explanation. One possibility might be a non-pericyclic reaction involving protonation at C11 of 7, hydride migration from C12 to the cation at C11m followed by deprotonation at C11.

Response: We agree with the reviewer. A [1,3]-antarafacial-hydrid shift is actually geometrically unfavorable, so we have modified it. According to a previous report (Tietze, L. F. & Grote, T. Synthesis of the reduced A-Unit (CI) of the antitumor antibiotic CC-1065. Chem. Ber. 1993, 126, 2733-2737. Fig. 1a), we describe this process as “isomerization”.

Figure 1. Analogous conversion of 8 to 7 in synthetic chemistry

We also believe that a reaction mechanism involving a carbocation intermediate is unlikely. We provide three reasons for this. First, if protonation occurs at C11 of 10 (reassigned in this revised submission) it will produce a primary carbocation at C11M; if protonation occurs at C11M of 10, it will produce a benzylic carbocation at C11. As far as we know, a primary carbocation is far less stable than a benzylic carbocation, so we think this kind of protonation is energetically unfavorable. Second, hydride migration from C12 to the cation at C11M is a cation-induced 1,3 hydride migration, and this process are usually realized by continuous 1,2 hydride migration. However, this mechanism is not consistent with our labeling experiment results using D₂O. And even if a direct migration of 1,3 hydrogen migration occurs, the proportion of such products will

be small. Third, according to a previous synthetic chemistry report (Yasuzawa, T. et al. Duocarmycins, potent antitumor antibiotics produced by *Streptomyces* sp. structures and chemistry. Chem. Pharm. Bull. 1995, 43, 378-391. Fig. 1b), treatment of a brominated compound with DBU (1,8-Diazabicyclo[5.4.0]undec-7-ene) in acetonitrile to produce a methylated product via dehydrobromination and isomerization. Since this reaction is carried out in the aprotic solvent, we speculate that there is no carbocation formation during the process.

The authors refer to an NMR spectrum supporting the labeling of 4 shown in Figure 4b. I cannot find this spectrum. What is the ratio of H vs D migration?

Response: We have provided this figure as Fig. 4b in this submission. In order to confirm the exact position of D incorporated into **D-7** (reassigned in this revised submission) isolated from enzymatic assays in D₂O, we compared the ¹H NMR of **7** produced in H₂O and the ²H NMR of **D-7** produced in D₂O, and found that the D in **D-7** either located in the C-12 or C-11M position. As for the ratio of H/D that migrates from C12 to C11M during the process of isomerization of **10** to **7**, it can be calculated from the peak area of C12-D and C11M-D in the ²H NMR of **D-7**. The ratio of the peak area of C12-D vs. C11M-D is about 3:1, which indicates that the H/D ratio that migrates from C12 to C11M is 3:1.

Page 7, Paragraph 1, line 6: The assignment of H138 as the base involved in phenol deprotonation is tentative. I suggest replacing “deprotonates the phenolic” with “is likely to deprotonate the phenolic”.

Response: We agree with the reviewer. We have revised related wording.

Page 7, Paragraph 1, line 5 (and several other places): I suggest replacing “grabs” with “abstracts”.
Page 8, Paragraph 2, line 5: replace “double” with “double bond”.

Response: We agree with the reviewer. We have improved our manuscript in this submission.

Page 8, Paragraph 2 line 9 from the end: I suggest replacing “which requires the phenolic hydroxyl and C-S bond about to break arranged collinearly on intermediate 5.” With “which requires that the HOMO of the phenol can overlap with the C-S s* orbital in a strained transition state”.

Response: We agree with the reviewer. This description is indeed more reasonable than that in our previous submission. However, we have omitted this information in this submission because we lack related supporting proofs.

Page 8, Paragraph 2 line 8 from the end: The authors state: “---whereas in the generation of off-pathway product 4, a more convenient E2 elimination promoted by a general base may occur much faster under the biochemical assay conditions *in vitro*.” This is not at all clear to me. In order for the E2 elimination to occur, the C11H and C-S bonds must be trans and coplanar. This requires a major movement of most of the atoms of SAM in the active site. This merits some discussion.

Response: We agree with the reviewer. We have provided a new figure as Fig. 5. We have made some modifications about related intermediates. Especially, we show the configuration of C11 in the intermediate **8** (reassigned in this revised submission), and in this configuration, the C11-C11M single bond can undergo free rotation to make it convenient for **8** to adopt a trans-coplanar conformation to undergo the E2 elimination to yield **10** with SAH as the co-product.

Page 10, Line 4: replace “centrifugalized” with “centrifuged”

Response: We agree with the reviewer. We have changed it.

Page 12, line 3: Please state that HPLC conditions are described in SI. Page S6: Please provide chromatography conditions for the purification of 3

Response: We agree with the reviewer. We have provided related information.

Page S10, line 4: Define DMPD.

Response: We agree with the reviewer. We have provided the full name for DMPD (N, N-dimethyl-p-phenylenediamine monohydrochloride) in this submission.

Page S30: Labels for the gene clusters are distorted. I recommend listing all 12 identified clusters.

Response: We agree with the reviewer. We have provided related information in this submission (Supplementary Fig. 25).

Thank you very much again for your consideration of our manuscript.

REVIEWERS' COMMENTS:

Reviewer #1 (Remarks to the Author):

The revisions the authors have made to this manuscript have addressed my earlier concerns. I support publication of this work in Nature Communications.

Reviewer #2 (Remarks to the Author):

The authors have effectively addressed reviewer concerns from the initial round of review. The introduction is now much more clear and provides a more interesting setup of the work. Overall, I think this is a significantly better manuscript than the original submission.

In the Discussion, the authors state that HemN-like proteins are distinct from other well-characterized radical SAM enzymes that ... cleaves its S-C bond to produce a dAdo radical. They should be careful with how they state this point, as their C10P also does this, it is just that the subsequent H-atom abstraction is from the methyl of another SAM, rather than another substrate. They might also want to point out that there are similarities to the radical SAM enzymes like RlmN and Cfr, except in these the methyl of one SAM is transferred first, prior to H-atom abstraction. The authors should also note that Dph2 is not a radical SAM enzyme, and so it is misleading to include it in this discussion without saying that it is not a radical SAM enzyme, or referring to it as a non-canonical radical SAM enzyme.

Minor points:

In Fig 2, it would be better to show the labels for chromatograms iv and v (and anywhere else the deletion mutants are used) in panel (a) as (Δ)c10P and (Δ)c10Q

In the following sentence, "factually" should be changed to "actually"
The very recent studies of the class C RSMTs TbtI, NosN, and ChuW have disclosed that they factually catalyze the transfer of a methylene from SAM to substrate³⁸⁻⁴².

In the sentence after the one above, I think that "believed" should be replaced by "thought of."
There are other grammatical issues in the Discussion as well, although I am not pointing all of them out here.

Reviewer #3 (Remarks to the Author):

The authors have addressed my questions and those of the other reviewers in a satisfactory manner and the paper is now suitable for publication.

Response letter for manuscript NCOMMS-18-00750A

Thank you very much for considering publication of our manuscript NCOMMS-18-00750A. We thank the reviewers again for their constructive suggestions. Based on their comments, we have improved the manuscript. Here we submitted the revised manuscript.

The following are our point-by-point responses.

Reviewers' comments:

Reviewer #1 (Remarks to the Author):

The revisions the authors have made to this manuscript have addressed my earlier concerns. I support publication of this work in Nature Communications.

Response: Thank the reviewer very much for the recommendation.

Reviewer #2 (Remarks to the Author):

The authors have effectively addressed reviewer concerns from the initial round of review. The introduction is now much more clear and provides a more interesting setup of the work. Overall, I think this is a significantly better manuscript than the original submission.

In the Discussion, the authors state that HemN-like proteins are distinct from other well-characterized radical SAM enzymes that ... cleaves its S-C bond to produce a dAdo radical. They should be careful with how they state this point, as their C10P also does this, it is just that the subsequent H-atom abstraction is from the methyl of another SAM, rather than another substrate. They might also want to point out that there are similarities to the radical SAM enzymes like RlmN and Cfr, except in these the methyl of one SAM is transferred first, prior to H-atom abstraction. The authors should also note that Dph2 is not a radical SAM enzyme, and so it is misleading to include it in this discussion without saying that it is not a radical SAM enzyme, or referring to it as a non-canonical radical SAM enzyme.

Response: We agree with the reviewer. We have rewritten the relevant discussion.

Minor points:

In Fig 2, it would be better to show the labels for chromatograms iv and v (and anywhere else the deletion mutants are used) in panel (a) as (Δ)c10P and (Δ)c10Q

Response: We agree with the reviewer. We have cited the relevant information accordingly.

In the following sentence, “factually” should be changed to “actually”

The very recent studies of the class C RSMTs TbtI, NosN, and ChuW have disclosed that they factually catalyze the transfer of a methylene from SAM to substrate³⁸⁻⁴².

Response: We agree with the reviewer. We have changed it.

In the sentence after the one above, I think that “believed” should be replaced by “thought of.” There are other grammatical issues in the Discussion as well, although I am not pointing all of them out here.

Response: We agree with the reviewer. We have changed it.

Reviewer #3 (Remarks to the Author):

The authors have addressed my questions and those of the other reviewers in a satisfactory manner and the paper is now suitable for publication.

Response: Thank the reviewer very much for the recommendation.